# mRNA-Based Therapeutics in Cancer Treatment

**DOI:** 10.3390/pharmaceutics15020622

**Published:** 2023-02-13

**Authors:** Han Sun, Yu Zhang, Ge Wang, Wen Yang, Yingjie Xu

**Affiliations:** 1Department of Biochemistry and Molecular Cell Biology, Shanghai Key Laboratory for Tumor Microenvironment and Inflammation, Shanghai Jiao Tong University School of Medicine, Shanghai 200025, China; 2Department of Oral Maxillofacial & Head and Neck Oncology, National Center of Stomatology, National Clinical Research Center for Oral Disease, Shanghai Ninth People’s Hospital, Shanghai Jiao Tong University School of Medicine, Shanghai 200011, China; 3Key Laboratory of Cell Differentiation and Apoptosis of Chinese Ministry of Education, Shanghai Jiao Tong University School of Medicine, Shanghai 200025, China

**Keywords:** messenger RNA, modification, delivery, mRNA vaccine, cancer immunotherapy

## Abstract

Over the past two decades, significant technological innovations have led to messenger RNA (mRNA) becoming a promising option for developing prophylactic and therapeutic vaccines, protein replacement therapies, and genome engineering. The success of the two COVID-19 mRNA vaccines has sparked new enthusiasm for other medical applications, particularly in cancer treatment. In vitro-transcribed (IVT) mRNAs are structurally designed to resemble naturally occurring mature mRNA. Delivery of IVT mRNA via delivery platforms such as lipid nanoparticles allows host cells to produce many copies of encoded proteins, which can serve as antigens to stimulate immune responses or as additional beneficial proteins for supplements. mRNA-based cancer therapeutics include mRNA cancer vaccines, mRNA encoding cytokines, chimeric antigen receptors, tumor suppressors, and other combination therapies. To better understand the current development and research status of mRNA therapies for cancer treatment, this review focused on the molecular design, delivery systems, and clinical indications of mRNA therapies in cancer.

## 1. Introduction

Messenger RNA (mRNA), a transient intermediator between genes and proteins, was first discovered in 1961 by Brenner et al. [1]. In 1990, in a study by Wolff et al. [2], in vitro-transcribed mRNA was directly injected into mouse muscles for the first time, and the corresponding protein products were detected, which proved the feasibility of mRNA therapy. Since then, several strategies have been explored to ameliorate the high immunogenicity and instability of IVT mRNA and its inefficiency in in vivo delivery. Advances in IVT mRNA with chemical modifications and in vivo delivery systems have expedited the development of mRNA as a new class of drugs. mRNA therapy has several advantages. The first is safety, as mRNA does not enter the nucleus; therefore, it has no risk of integration into the genome. Second, mRNA can be degraded through normal cellular pathways, and the metabolites are natural. Third, for any target protein of a known sequence, mRNA can be quickly produced in vitro by an enzymatic reaction, thereby avoiding complex manufacturing [3].

The effectiveness of these two COVID-19 mRNA vaccines in the real world has again aroused an upsurge in mRNA therapy research worldwide. In cancer treatment, in 2017, Sahin et al. [4] reported for the first time a clinical trial of applying personalized mRNA cancer vaccines against multiple antigens to patients with melanoma. In addition to mRNA cancer vaccines, treatment methods such as mRNA encoding immunomodulatory factors, tumor suppressor genes, and antibodies are also in the preclinical/clinical stages of cancer treatment. This review focused on the modification regulation/sequence optimization system of in vitro transcription of mRNA molecules, in vivo delivery, and the clinical application scenarios of mRNA as a promising new generation of biomacromolecule drugs in the field of cancer.

## 2. Components and Design of IVT mRNAs

Similar to natural mRNA, IVT mRNA contains five components from the 5′ to the 3′ end: a 5′ cap, 5′ untranslated region (5′ UTR), coding sequence (CDS), 3′ UTR, and poly(A) tail (Figure 1). IVT mRNAs are susceptible to non-enzymatic decay and attack by the host cellular mRNA degradation system [5]. They should be optimized to take better advantage of the eukaryotic protein translation machinery than endogenous mRNAs and to ensure sufficient half-life to achieve protein expression levels and durations. With a deeper understanding of the influence of mRNA sequence and structure on its biological performance, the emergence of diverse design strategies provides broader options for mRNA optimization, typically with chemical synthesis, computational biology, and bioinformatics. The current advances in the design of the five elements of IVT mRNA are discussed below.

### 2.1. 5′ Cap

In eukaryotic transcription, 7-methylguanosine (m^7^G) is connected to the 5′ terminal nucleotide of mRNA via a 5′-5′ triphosphate bond to form the 5′-cap structure (m^7^G(5′)ppp(5′)Np). The cap structure recruits eukaryotic initiation factor 4E (eIF4E) to initiate the translation process [6]. In addition to facilitating polyadenylation at the 3′ end [7,8,9], splicing [10,11], and nuclear export of endogenous precursor mRNA (pre-mRNA) [11,12] in natural situations, the cap structure in the case of IVT mRNA is essential for translation efficiency and stability (half-life). Based on the number of methylated nucleotides at the C2′ position (2′-OMe) of the ribose from the 5′ end, cap 0 (m^7^GpppNp) can be distinguished from cap 1 (m^7^GpppNmpNp) or cap 2 (m^7^GpppNmpNmp) by the ribose methylation at C2′ or both C2′ and C3′, respectively (Figure 2a). Uncapped mRNAs or those with cap 0 can be recognized by pattern recognition receptors (PRRs), such as retinoic-acid-inducible gene I (RIG-I) and melanoma-differentiation-associated protein 5 (MDA5), triggering the IFN innate immune response to inhibit translation and protein synthesis [13].

There are two common methods for IVT mRNA capping: one is the addition of cap analogs as RNA polymerase substrates for one-step co-transcription capping, and the other is to employ specialized capping enzymes for post-transcription capping. Co-transcription capping with synthetic cap analogs, such as anti-reverse cap analogs (ARCAs) with a typical structure of m_2_^7^′^, 3^′^−O^GpppG or m^7^(3)dGpppG (Figure 2a), is used earlier and more frequently, allowing IVT to be completed in a single step. In ARCAs, the C3′ hydroxyl of m^7^G is modified such that the 3′-5′ phosphodiester bond cannot be formed, ensuring correct capping. Modification of the C2′ position can also prevent reverse capping [14]. Linking two nucleosides with tetra- instead of triphosphate and appropriate modification of this oligophosphate chain can improve stability and translation efficiency by facilitating eIF4E binding [15,16,17,18,19] (Figure 2b). At present, CleanCap^®^, a next-generation one-pot capping technology, enables co-transcription capping without cap analogs with a capping rate of >95% [20]. Post-transcription capping implies that an additional enzymatic capping step is performed after transcription. At present, the vaccinia capping enzyme (VCE) is commercially available and completes mRNA capping similarly to the eukaryotic capping machinery to produce IVT mRNA with cap 0 (m^7^GpppNp). Commercialization of the 2′-O-methyltransferase VP39 in vaccinia viruses for further processing renders IVT mRNA with a cap 1 [21]. Some viral enzymes, such as VP4 in Bluetongue viruses, can obtain cap 1 directly [22]. Capping, as an additional step, complicates industrial production and may reduce yield.

### 2.2. Untranslated Region (UTR)

There are many cis-elements and secondary structures on the 5′ and 3′ UTRs of eukaryotic mRNA that interact with various RNA-binding proteins (RBPs), and it is widely accepted that 5′ and 3′ UTRs are essential for mRNA translation and stabilization. However, due to the lack of in-depth insight into the regulatory mechanisms involved, current IVT mRNA design predominantly uses UTRs derived from genes with high expression, such as α-globin or β-globin UTRs from *Xenopus laevis* and *Homo sapiens* [23,24], which leaves much room for optimization. 

Several aspects are relevant to UTR sequence engineering. (1) Structure: in general, translation efficiency suffers when excess secondary structures are introduced into the 5′ UTR [25]. (2) Kozak sequence: For efficient translation of the CDS region, the “AUG background” nucleotides around the initiation site AUG should be an optimal Kozak sequence. In vertebrates, the general sequence can be described as follows: (gcc)gccRccAUGG (AUG: initiation codon; [gcc]: importance unknown; R: purine (adenine accounts for ~97%); lower case: most common bases (may vary); upper case: highly conserved bases) [26]. (3) Upstream start codon (upstream AUG, uAUG) or upstream open reading frames (uORFs): unexpected start codons or open reading frames upstream of the designed CDS may interfere with target protein expression [27] or cause the regulator of nonsense transcript 1 (UPF1)-dependent degradation of mRNA [28,29,30]. (4) Internal ribosome entry sites (IRESs): The IRES consists of hundreds of nucleotides (nt) that were first discovered in viral gene expression and later confirmed in some mRNAs of eukaryotic cells. It can bind to the 40S subunit of the ribosome and initiate translation of CDS, bypassing canonical cap-dependent translation initiation. The IRES in the 5′ UTR is approved for the translation of IVT mRNA without cap [31,32]. 

Optimization of the 3′ UTR may improve IVT mRNA stability [33]. Adenylate–uridylate-rich elements (AREs; usually, AUUUA motifs are scattered or overlapped within or near U-rich regions) are the most common mRNA stability determinants in mammals. Some AU-rich element RNA-binding proteins (AUBPs) stabilize mRNA, whereas others, such as AU-rich element RNA-binding protein (AUF1), tristetraprolin (TTP), and human antigen R (HuR), form exosomes that mediate 3′→5′ degradation starting from the poly(A) tail [34,35]. Guanosine–uridine-rich elements (GREs) [36] and cytosine-rich elements (CREs) also regulate mRNA expression through combined factors. The 3′ UTR also plays a role in mRNA transport and localization. At specific regions in the 3′ UTR, mRNA localization signals function as zip codes and interact with trans-acting factors in a sequence- and structure-dependent manner to determine the subcellular transport and anchoring of the mRNA, which affects translation. For IVT mRNA sequence engineering, it is advisable to locate the best translation site based on the synthesis, processing, (secretion), and functional pathway of the encoded protein by editing the 3′ UTR [37].

Machine learning combined with high-throughput sequence screening may be a hotspot for next-generation mRNA design. Castillo-Hair et al. [38] conceived a three-step workflow for this pattern, which can be interpreted as follows: establishment of a training set from experimental data, generation and verification of a machine learning model, and model-based computational mRNA design. The first step requires a massively parallel screening assay of the mRNA library containing the sequence fragment to be optimized with freely chosen metrics to quantify mRNA performance. The experimental results in step three can serve as data input for feedback [33,39,40,41]. Since plasmid-based screening cannot reliably characterize the expression level of IVT mRNA [42], a well-designed screening assay is required [41]. The machine-learning-based framework is expected to thrive in future mRNA design because of its iterative optimization capability and open-source linkage of data and algorithms.

### 2.3. Coding Sequence (CDS)

The coding sequence (CDS) is the core of the IVT mRNA. Codon optimization in the CDS region refers to synonymous substitutions that regulate protein translation while avoiding attack by endonucleases [43]. Considering the “codon bias,” one common method is to use more frequently used codons in human mRNAs as replacements for rare codons to accelerate mRNA translation and avoid degradation caused by translation blockage [44,45]. In addition, the frequency of certain codon pairs (“codon pair bias”) and dinucleotides is thought to be related to the translation efficiency [46,47,48]. Optimization of G/C at the third position of the codon stabilizes mRNA and promotes translation. Hia et al. [49] combined high-throughput sample analysis and experimental methods to comprehensively explain that GC3 (G or C at the third position of codons) and the GC content of the whole mRNA affect stability. Since the exchange of G↔A and C↔U at position three of the codon has little effect on the determination of amino acids, GC3 and AU3 content can be used to some extent as markers to reflect the properties of mRNAs. Studies have shown that GC3-rich mRNAs have a higher ribosome reading rate and protein expression efficiency than AU3-rich mRNAs. Total GC content provides a translation-independent stabilization effect that correlates with mRNA degradation mediated by RBPs [50]. Moreover, decreasing the U ratio attenuates the activation of PRRs such as Toll-like receptor-7 (TLR-7), resulting in mRNA degradation [51,52,53,54,55].

Both the primary and high-level structures are essential for CDS translation. Undesirable codon optimization may generate unanticipated secondary structures and unfavorably affect the kinetics and authenticity of ribosome scanning, resulting in erroneous wobble pairing and reduced quality and quantity of protein expression [56,57,58]. Therefore, the optimization of both mRNA and secondary structures may have a synergistic effect in terms of enhancing and prolonging protein expression.

Computer-aided mRNA design is applicable to CDS engineering. In addition to the machine learning framework mentioned above, LinearDesign, a word lattice parsing-based algorithm developed by Baidu that enables the rapid optimization of sequence design, is promising for mRNA-based cancer treatment [59].

### 2.4. Poly(A) Tailing

Polyadenylation of pre-mRNA in eukaryotic cells occurs after transcription and before transport from the nucleus to the cytoplasm to form the poly(A) tail, which contains consecutive adenine nucleotides (50–250 nt) bound by PABPs to promote nuclear export [60], increase translation [61,62], and inhibit degradation [63]. In the closed-loop model of mRNA translation, poly(A) prevents mRNA degeneration [64]. Poly(A)-specific ribonucleases (PARNs) with 3′→5′ exonuclease activity can bind to the 5′ cap to cause deadenylation [65], which acts as an important initiator of some crucial mRNA decay pathways. In eukaryotes, the poly(A) of most cytoplasmic mRNAs gradually becomes shorter, and mRNAs with shorter poly(A) tails have been found to be less translated and more rapidly degraded [66]. However, with advances in poly(A) tail sequencing, seemingly paradoxically, tails truncated to a minimal length are emerging as a feature of highly enriched and well-expressed transcripts, which may be partly due to the shrinking of binding regions for deadenylases and translation inhibitors [67].

IVT mRNAs with different poly(A) tail lengths, ranging from 60–70 nt [68] to the generally accepted appropriate length of 120–150 nt, have been tested in various cell lines [69,70]. The optimal tail length of IVT mRNAs requires adaptation to a specific case, based on the intrinsic properties of IVT mRNAs and the cytoplasmic environment [71]. 

There are two methods for synthesizing IVT mRNAs with poly (A) tails of a specific length. One is to insert the poly[d(A/T)] sequence of a certain length into the DNA template (pDNA or PCR product), and the other is to perform post-transcriptional enzymatic polyadenylation with recombinant poly(A) polymerases. The former provides mRNA with a specific length of poly(A) tail in one step, which facilitates the quality control of IVT mRNA, particularly for clinical applications. The latter offers the possibility of inserting chemically modified nucleotides into the poly(A) tail to increase its stability and promote translation [72]. A conserved hairpin, rather than the poly(A) tail at the 3′ end of histone mRNAs, performs a poly(A)-like function that can replace poly(A) or be appended downstream as a possible optimization [73,74,75].

### 2.5. Chemical Modification

The role of nucleotide modification of mRNA is multifaceted and context-dependent [76]. It affects dsRNA formation in IVT, secondary structure, translation, and immunogenicity [76,77,78,79]. As foreign substances with immunostimulatory properties, unmodified IVT mRNAs can be recognized by the PRRs of innate immune cells, eventually leading to the breakdown of the entire translational apparatus of the host cell. The ability to elicit an innate immune response is also referred to as intrinsic adjuvant activity. Appropriate chemical modifications can reduce the degradation of mRNA and maintain the stability of its secondary structure [78]. More importantly, chemical modification of nucleotides can regulate the immunogenicity of IVT mRNA.

In tumor IVT mRNA vaccines, overdue intrinsic adjuvant activity is unfavorable for antigen processing and presentation and insufficient for T and B cell activation; however, the moderate intrinsic adjuvant activity of IVT mRNA vaccines can promote the maturation of antigen-presenting cells (APCs), such as DCs (cytokines TNF, IL-12, and IL-6 involved), to exert APC function in adaptive immunity [80]. According to our cancer therapy strategy, methods for IVT mRNA optimization, including chemical modification and purification techniques such as high-performance liquid chromatography (HPLC) [81], should be applied to adjust immunogenicity to a suitable level.

Modification of nucleosides prevents the recognition of PRRs, including TLR3, TLR7, and TLR8 [82]. Adenosine is commonly replaced by *N*^6^-methyladenosine (m^6^A) or *N*^1^-methyladenosine (m^1^A), cytidine by 5-methylcytidine (m^5^C) or 5-hydorxymethylcytidine (hm^5^C), and uridine by pseudouridine (ψ), 2-thiouridine (s^2^U), *N*^1^-methylpseudouridine (m^1^ψ), or 5-methyluridine (m^5^U) (Figure 3). Karikó et al. [82,83] found that m^5^C and ψ significantly reduce the immune response and improve translation efficiency. In a study by Kormann et al. [84], 14 days after intramuscular administration of IVT mRNA containing 25% uridine and cytidine replaced with s^2^U and m^5^C, respectively, protein expression levels were 4.8- and 4.4-fold higher, respectively, than those in mice injected with unmodified mRNA. As the modification in two FDA-approved COVID-19 mRNA vaccines, m^1^ψ reduces immunogenicity compared with canonical U, with the change in mRNA structure affecting translation initiation and half-life [77,78,79,85]. In general, the proportion of chemically modified nucleotides is determined by their loading ratio to achieve optimal protein expression [84,86]; post-transcriptional nucleotide modification could also be an alternative [87].

## 3. mRNA Delivery System

Although the first attempt to inject naked mRNA has been shown to generate an encoded protein [2], the lack of an efficient delivery system has limited its use in the early years. Naked mRNAs can be rapidly degraded by extracellular RNases and have difficulty passing freely across cell membranes. Thus, to play an important role in vivo, mRNA delivery platforms must overcome several extracellular and intracellular barriers. These include protection from degradation by nucleases in physiological fluids, protection from interception by the mononuclear phagocytosis system, elimination of glomerular filtration after systemic administration, and improvement in the ability of mRNA to reach the cytoplasm for translation after reaching the target tissue and being swallowed by the target cell. Therefore, appropriate methods are required for efficient mRNA delivery to achieve these functions. After being encapsulated in a delivery vehicle, mRNA is able to enter the target cells through multiple mechanisms, which depends on the properties of the delivery platform and the cell type. For instance, mRNA delivered by lipid nanoparticles (LNPs) can be internalized by micropinocytosis and endocytosis; polyplexes enter the cells via caveolae-mediated endocytosis while lipoplexs via clathrin-mediated endocytosis or fusion with the cell membrane [88]. The optimization of mRNA delivery systems is of great importance for mRNA drug development.

### 3.1. Naked mRNA Injection or Electroporation

Naked mRNA can hardly enter cell lines cultured in vitro because of its negative charge and large size [89]. The in vivo delivery efficiency of naked mRNA is highly dependent on the route of administration. Naked mRNA has low delivery efficiency, except with subcutaneous injection [89]. Eukaryotic cells can take up naked mRNA; however, the uptake efficiency of naked mRNA is too low (<1%) to have a significant effect on most somatic cells, except dendritic cells (DCs). Some researchers have suggested that immature DCs in the lymph nodes or dermis can selectively take up naked mRNA via micropinocytosis [90,91]. Thus, the injection of naked mRNA is now used for vaccines containing encoded antigens, mainly intradermal [92,93] (i.d.) or intranodal [94,95] injections. Some adjuvants are added to enhance the therapeutic immune response triggered by naked mRNA, such as granulocyte–macrophage colony-stimulating factor (GM-CSF) [93] or tyrosine kinase 3 (FLT3) ligand [96]. Naked mRNA administration has made some progress in both cancer vaccines [4] and infectious disease vaccines [97].

Some ex vivo loading methods have been used for immunological application of mRNA. Although DCs have been shown to engulf naked mRNA in vivo, ex vivo transfection still has higher efficiency and specificity. In these cases, mRNA is introduced by electroporation, forming membrane pores and directly entering the cytoplasm. The ex vivo mRNA loading strategy was applied to DCs [98] and CAR-T cells [99,100]. The ex vivo procedure is usually associated with superfluous costs and risks, but this shortcoming no longer exists with CAR-T, which initially involves an ex vivo procedure. In vivo electroporation has been used in preclinical studies to increase uptake efficiency [101,102].

### 3.2. Liposome and RNA Lipoplexes (LPX)

Lipids have always been attractive materials for mRNA transfer because they are selectively electrical and biodegradable. Among them, cationic lipids are the first choice for nucleic acid delivery because they carry an electrical charge opposite to that of RNA. Liposomes are closed spherical lipid bilayers that form an internal cavity that can hold aqueous solutions. Liposomes usually contain cationic lipids (DOTAP [103] and DOTMA [104]) that can bind negatively charged RNA and some helper lipids (DOPC, DOPE, and DSPC) to form the lipid bilayer structure.

RNA LPX refers to cationic liposomes that are mixed with RNA. Liposomes are lipid carriers of nucleic acids that are used as transfection reagents in vitro. Owing to their positive surface charge, they can form complexes (spherical or continuous bilayer structures) with negatively charged nucleic acids. LPX was first used for RNA delivery in 1989, inspired by its success in DNA delivery [23], and has now been widely used for mRNA vaccine delivery. RNA-LPX was prepared by diluting the RNA with liposomes in ethanol and sodium chloride solutions at a selected charge ratio [105].

Changing the ratio of cationic lipids to RNA could alter target specificity. Decreasing cationic lipid content could allow systematic delivery to the spleen. This improved mRNA-LPX, when injected intravenously, could systematically be targeted to the spleen DCs and serve as a cancer vaccine [105]. The same formulation is effective in the treatment of autoimmune encephalomyelitis by delivery of autoantigens [106] and in the treatment of melanoma by delivery of tumor-associated antigens (TAAs) [107]. LPX mRNA can be administered via several routes, with intravenous administration being the most common. Intravenous administration showed the highest delivery efficiency and satisfactory immune organ specificity. LPX mRNA is mainly used for the systematic delivery of cancer vaccines, and several projects are in clinical trials (summarized in another review [108]).

### 3.3. Lipid Nanoparticle (LNP)

At present, the LNP-based mRNA delivery system is considered to be one of the most advanced and promising delivery systems, particularly after the great progress achieved with regard to the COVID-19 vaccine. LNPs differ from liposomes in that they have micellar structures within their particle core [109]. 

LNPs typically contain four types of lipid materials: cationic or ionizable lipids, cholesterol, phospholipids, and polyethylene glycol (PEG) lipids. Of these, cationic/ionizable lipids play an important role in forming a core with mRNA through electrostatic interactions to protect against RNase. LNPs are prepared via self-assembly and rapid mixing, which is generally facilitated by microfluidic chip devices [110].

Initially, cationic lipids, such as DOTMA/DOTAP, were used because they not only bind anionic mRNA but also fuse the membrane to promote cellular uptake and endosomal escape. Although cationic lipids showed promising effects for successful delivery, the permanent charge they carried resulted in high cytotoxicity and limited their potential applications [111]. Ionizable lipids, which have no charge at physiological pH but become positively charged at low pH, were introduced as second-generation cationic lipids to reduce cytotoxicity. In the preparation of LNPs, ionizable lipids carry a positive charge and efficiently bind mRNA. This type of ionizable LNP remained neutral in a physiological pH environment, thus reducing cytotoxicity. In addition, endocytosis of LNPs could trigger a pH decrease mediated by the proton pump, allowing them to escape more easily [112,113]. Hundreds of ionizable lipids have been developed for various applications. Several parameters, such as tail length, unsaturation, branching, and pK_a_, significantly influence the properties of ionizable lipids. The unsaturation and tail length of ionizable lipids affect the pK_a_, fusogenicity, cellular uptake, and delivery efficiency [114]. Unsaturated ionizable lipids with proper unsaturation have higher efficiency in mRNA delivery [115,116]. pK_a_ is thought to affect the particle characteristics, cellular uptake, and endosomal escape. Some studies have shown that the optimal pK_a_ value for mRNA delivery to the liver is 6.2–6.8; however, it varies from organ to organ [117,118]. Multi-tailed [119] and branched-tail [120] ionizable lipids have also been developed and have progressed in their expression. Significant efforts have been made to optimize these properties to achieve the best delivery efficiency and lowest cytotoxicity. The nitrogen-to-phosphate (N/P) ratio is a major property in LNP packaging formulations, and most formulas adopt a ratio of 6 or 3. DLin-MC3-DMA, which was optimized from DODMA, DLin-DMA, and DLin-KC2-DMA [121], is a promising product for use in commercial siRNA drugs [122] and various mRNA applications [123,124,125]. ALC-0315 and SM-102 were used by BioNTech and Moderna, respectively, for the COVID-19 vaccine [126] (Figure 4). Many companies have developed various ionizable compounds, such as LP01 (Intellia Therapeutics) [127] and ATX (LUNAR composition, Arcturus) [125], which have been described in detail in another review [128].

Helper lipids, including phospholipids and cholesterol, are incorporated to promote formulation stability, membrane fusion, and escape from the endosome [129]. For example, 1,2-distearoyl-sn-glycero-3-phosphocholine (DSPC) is a widely used helper phospholipid that can stabilize the structure of LNPs [130]. Both the mRNA-1273 [131] and BNT162b2 [132] COVID-19 vaccines use DSPC. DOPE is another commonly used phospholipid. Some studies show that an optimized formulation containing DOPE instead of DSPC could destabilize endosomal membranes to promote endosomal escape [119]. Hydrophobic and rigid cholesterol can fill the gaps in lipid membranes to stabilize the structure, as it plays a role in the cytoplasmic membrane [133]. PEG is normally anchored to lipids to prolong their half-life in the circulation [130] by decreasing macrophage-mediated clearance [134] and apolipoprotein adhesion. In addition, modification of PEG could improve steric stability to extend storage time [135]. DMG-PEG is the most commonly used agent, whereas some formulations contain DSPE-PEG to further extend the half-life in the circulation. DSPE-PEG has saturated alkyl chains (C18) in contrast to DMG-PEG (C14). Longer alkyl chains affect the efficiency of cellular uptake and endosomal escape but dissociate more slowly in the circulatory system [136]. The amount of incorporated PEG lipid determines the particle size [137].

In addition to their successful application in the local delivery of vaccines, LNPs have also enabled targeted delivery to organs via systematic administration [138,139]. This specificity of organ selectivity is mainly based on the global/obvious pK_a_ of the lipid and serum protein interactions of selective organ-targeting (SORT) nanoparticles [140]. The alkyl length of the lipid can also alter delivery to the target organs, such as the liver and spleen [141]. Neurotransmitter-derived lipidoids (NT-lipidoids) have also been developed to transport mRNA cargo across the blood–brain barrier (BBB). Cell-targeted delivery can also be achieved by decoration with targeted antibodies [142,143]. This type of LNP-targeting T cell has been shown to produce CAR-T cells in vivo by targeted delivery of mRNA encoding CAR [144]. The formulation could also be modified by adding new components to improve performance, for example, the addition of poly (disulfide amide) (PDSA) to promote triggered release in cancer [145,146].

In addition to release specificity, release efficiency is another important concern for LNPs. Lipid materials induce efficient cellular uptake; however, very few mRNA transporters can escape from endosomes and enter the cytosol. Studies on siRNA [147] and mRNA [117] revealed that less than 5% could escape from the endosome because the RNA could only be released within a limited time window [147]. Although many efforts have been made to elucidate the mechanism of endosomal escape, it remains unclear to date.

Several quality control standards have been established to ensure the safety and efficacy of the mRNA-LNP products. The encapsulation efficiency and concentration of mRNA are usually determined using the Quant-iT RiboGreen RNA assay [148]. The physicochemical properties of the particles, including particle size distribution, polymer dispersity index (PDI), and zeta potential, were measured using particle size analyzers. The morphology, size, and structure of the nanoparticles were visualized using transmission electron microscopy (TEM). Intraserum stability [149], anti-RNase performance, and storage stability [150] were also evaluated to ensure a stable product quality. The payload distribution and capacity of mRNA-LNPs are critical but remain a challenge, and some researchers have developed a method based on multi-laser cylindrical illumination confocal spectroscopy (CICS) [151].

### 3.4. Polymer-Based Nanoparticles

Polymeric materials, such as polyetherimide (PEI), poly-L-lysine (PLL), polyamidoamine (PAMAM), and poly (lactic-co-glycolic acid) (PLGA), are other options that are not as clinically advanced as lipids (Figure 5). Polymeric materials have the advantages of easy synthesis and scalability. However, they have some disadvantages compared to lipids, such as polydispersity and biodegradability. To overcome the shortcomings of biodegradability, various efforts have been made to extend branching structures [152] or construct biodegradable domains [153].

PEI is one of the most widely used polymeric materials, and its linear derivative is commercially available and can be used for mRNA delivery [154]. However, PEI is still highly toxic because commercial high-molecular-weight PEI is not degradable [155]. Some researchers have tried to solve this problem by adding acid-labile imine linkers [155] or by reducing its molecular weight and introducing branching [156].

Most polymeric materials are cationic; however, PLGA is anionic and has received FDA approval for certain applications. Anionic polymers are unlikely to bind to negatively charged mRNA, so they are always used in conjunction with other cationic polymers. PLGA improves delivery efficiency [157] and prolongs the half-life of the mRNA complex in the circulatory system [158,159]. By coupling an anti-CD8 antibody to another anionic material, polyglutamic acid (PGA), nanoparticles of poly (β-amino ester) (PbAE)/PGA-anti-CD8 were sufficient to target circulating T cells. mRNA could be delivered to T cells by injection, and encoded CARs or TCRs could mediate therapeutic effects [160].

### 3.5. Cationic Nanoemulsions (CNEs)

CNEs are oil-phase dispersions in the aqueous phase. CNEs are mainly composed of two parts: one is a cationic lipid such as DOTAP, which can be added to the oil phase to bind mRNA; the other is an oil-in-water emulation adjuvant composed of squalene and surfactants. CNEs are prepared by different strategies, in which the aqueous phase containing buffer and Tween is mixed with the oil phase containing cationic lipid, Span 85, and squalene [161]. It has been used mainly for DNA transport and as a self-amplifying mRNA vaccine with valid efficacy [162].

### 3.6. Protamine-Based Delivery 

In addition to cationic lipids, negatively charged mRNA can also be transferred by cationic peptides via electrostatic interactions. Similar to lipids, the amount incorporated into the complex and the expression efficiency of charged mRNA are determined by the N/P ratio [163]. Protamine is one of the best-known cationic peptides and was used in early studies. Protamine and mRNA can form condensed nanoparticles that protect mRNA from degradation by serum RNase [164,165]; however, this overly tight combination could also impair expression efficiency [166]. Protamine-formulated mRNA elicited a stronger immune response than naked mRNA [165] but could be advantageous in the application of mRNA vaccines [167,168,169].

## 4. mRNA-Based Cancer Immunotherapies

Immunotherapies are used to eliminate cancer cells by activating the innate and adaptive immunity, and various studies and strategies have been well tested. Owing to their efficacy and wide range of applications, immunotherapies are considered the most promising strategies for cancer treatment. There are a number of stepwise processes in the immune response to cancer. First, cancer cells release antigens that are taken up by DCs, presented on the major histocompatibility complex (MHC), and recognized by T cells to initiate proliferation and activation. Effector T cells then recognize and kill cancer target cells via T cell receptors (TCRs) and peptide–MHC-specific recognition, releasing more tumor antigens to expand the immune response. However, there are many reasons for the poor performance of the autoimmune response in patients with cancer. For example, (1) a low abundance of tumor antigens protects them from being presented by MHC; (2) DCs and T cells cannot recognize the antigens due to peripheral tolerance; (3) immunosuppression may be caused by the tumor microenvironment; and (4) immune suppression markers on the surface of cancer cells assist cancer cells in causing immune suppression [170,171]. To restore and strengthen cancer immunity, cancer immunotherapies that target different targets have been developed. Among these, cancer vaccines based on mRNA platforms have been rapidly developed. In particular, the recent FDA approval of two mRNA-LNP vaccines for COVID-19 prevention makes the clinical use of mRNA vaccines in cancer treatment promising. In this review, we present the preclinical/clinical cases, characteristics, and prospects of mRNA-based cancer immunotherapies. Based on the different mechanisms of immunotherapies mediated by mRNA [172], mRNAs can be divided into (1) neoantigen mRNA, (2) tumor-associated antigen (TAA) mRNA, (3) antibody mRNA, and (4) immunomodulator mRNA (Figure 6).

### 4.1. Neoantigen mRNA Vaccines

The most critical aspect of tumor antigen mRNA vaccine design is the selection of antigens that are ideally expressed only by cancer cells and are immunogenic [173]. This class of cancer cell epitopes is referred to as neoantigens or tumor-specific antigens (TSAs). In non-viral pathogenic human tumors, new epitopes are exclusively generated by tumor-specific DNA alterations caused by genetic instability [174]. These DNA changes include nonsynonymous mutations, frameshift mutations (insertions or deletions), gene fusions, post-translational modifications that alter the amino acid sequence, and intron retention [175,176,177,178,179]. In addition, post-translational modifications that alter the amino acid sequence and intron retention at the mRNA level can lead to the expression of non-autologous proteins. These new epitopes with individual specificity, called neoantigens [180], enable the immune system to recognize and destroy a tumor carrying these mutations. Epitopes from viral open reading frames (ORF) also contribute to neoantigens in virus-associated tumors, such as those caused by human papillomavirus (HPV). Thereafter, they undergo cytosolic degradation, are processed into short peptides (8–10 amino acid residues), and then transported to the endoplasmic reticulum to be loaded onto HLA molecules [181]. In contrast to autologous sequences, to which the immune system is tolerant, the ‘foreign’ peptide will be recognized by the T cell receptor (TCR) of CD8^+^ T cells and activated cytotoxic T lymphocytes (CTLs), which are responsible for the killing of tumor cells [182,183]. However, although tumor cells have many mutations, few are recognized by the patient’s own T cells, because neoantigen-specific T cell reactivity is generally limited to a few mutant epitopes [184]. One way to break the immune tolerance of T cells is to use mRNA to express neoantigen peptides to establish systemic DC targeting and neoantigen-specific T cell immunity.

Some neoantigens have high prevalence and conserved mutation profiles and are referred to as shared neoantigens, which have significant potential for use as broad-spectrum therapeutic cancer vaccines for patients with the same mutated genes. When the same neoantigen is present in a patient’s tumor cell, the corresponding off-the-shelf neoantigen-targeted immunotherapy can be used for treatment, which can significantly shorten the development cycle. For example, BRAF V600E, ERBB2 S310F, KRAS G12D, PIK3CA E545K, etc., are all generated by somatic mutations that are common in cancer patients [185]. Approaches to predict and prioritize immunogenic shared neoantigens are becoming more comprehensive, opening up new opportunities to develop neoantigen-targeted therapies in a very general way. For example, researchers have used computational epitope prediction, biochemical analysis, and proteomic analysis to predict and identify an mKRAS G12 peptide with high stability and affinity to HLA-A and HLA-B in a specific race [186]. In 2018, Moderna and Merck developed a novel shared antigen mRNA vaccine formulated with lipid nanoparticles called V941 (mRNA-5671), which targets the four most common KRAS mutations (*G12D*, *G12V*, *G13D*, and *G12C*) in solid tumors. Preclinical data show that after vaccination in a mouse model, the neoantigen is translated to induce CD8^+^IFN^+^ T cells that specifically target KRAS mutant tumor cells. Phase I trials of mRNA-5671 were recently completed in two groups (NCT03948763), either as monotherapy (intramuscular injection) or in combination with the anti-PD-1 antibody pembrolizumab (intravenous injection) to assess the safety and tolerability, involving 100 patients with lung, pancreatic, and colorectal cancers (not published yet). Since all types of HPV encode “early proteins” (E proteins: E1, E2, E6, E7) and “late proteins” (L proteins: L1, L2), the development of mRNA vaccines for HPV-positive malignancies has also evolved rapidly [187]. BNT113 (HPV16 E7 mRNA), an intravenous cancer vaccine that efficiently matures and amplifies antigen-specific effector and memory CD8^+^ T cells, was tested in mice using lipoplex (LPX) delivery. Its administration mediated tumor regression and prevented tumor recurrence in two HPV-positive mouse tumor models (TC-1 and C3) and showed a combined effect with PD-L1 inhibitors [188]. BNT113 in combination with anti-CD40 (HARE-40) is currently being tested in a phase I/II vaccine dose-escalation study in patients with advanced HPV16^+^ cancer (NCT03418480). Another phase II trial of BNT113 combined with pembrolizumab versus pembrolizumab alone as a first-line treatment in patients with HPV16^+^ head and neck cancer expressing PD-L1 is also underway (NCT04534205).

However, most cancer mutations are unique to each individual patient and require a personalized medical approach; thus, a highly specific procedure has been developed. Surgically resected tumors, tumor biopsies, and healthy blood cells were collected, and the extracted DNA from the samples was subjected to whole-exome and RNA sequencing to identify nonsynonymous mutations. Whether a mutation can be used as a therapeutic target depends on several critical factors [189]: (1) the mutated sequence can be translated into a protein in tumor cells, and the expression level of the originating gene should be greater than 33 TPM; (2) the mutated protein can be processed into a peptide; immunogenic peptides usually have low hydrophobicity and mutations do not occur at the second amino acid site; (3) the peptide can be presented by MHC with a binding stability greater than 1.4 h; (4) the mutated peptide has high affinity, which is usually stronger than 34 nM for MHC molecules; and (5) the mutated peptide–MHC complex has high affinity, ranging from 30 nM to 26 pM, for the T cell receptor (TCR) [190]. Therefore, the prediction of neoantigens requires not only the identification of mutations expressed in the genome, but also data on the patient’s MHC type [191]. A number of computational, biochemical, proteomic, and immunological assays have been used to predict the high affinity, immunogenicity, and expression efficiency of mutant peptides and HLA in tumors. Furthermore, a number of MS-based immunopeptidomic datasets such as IEDB [192], SysteMHC Atlas [193], and PRIDE [194] have been used in machine learning for neoantigen prediction. Tools such as NetMHC [195], MHCflurry [196], NetMHCpan [197], PSSMHCpan antigen-garnish [198], pVAC-Seq [199], and others have been widely used to predict peptide–HLA affinity based on various algorithms. The expression of mutated alleles and the processing and presentation of neoantigens can be confirmed by RNA-seq [200], HLA immunoprecipitation, and targeted mass spectrometry separately [201]. However, candidate neoantigens selected on a computer may not be recognized by T cells; therefore, it is necessary to verify the presentation and immunogenicity of neoepitopes [202,203]. Biochemical assays were performed to characterize the affinity and stability of peptide–HLA (p-HLA). Immunological datasets were collected by co-culturing T cells with mature dendritic cells (mDCs) pulsed with candidate epitopes [186,200] or stimulating peripheral blood mononuclear cells (PBMCs) from patients with neoepitopes, followed by T cell activation assessment by IFN-γ-ELISPOT, flow cytometry, etc. [204]. Subsequently, a series of potentially immunogenic peptides can be selected based on their immunogenicity and protein-binding affinity. 

A one-step procedure for the design and synthesis of neoantigenic mRNA has been developed. A patient-tailored DNA plasmid encoding a selected set of several neoantigens in tandem with minigenes (TMG^NEO^ plasmid) was developed (Figure 7). It has been reported that the combination of an N-terminal leader peptide with MITD bound to the C-terminus of the antigen significantly improves the presentation of HLA epitopes in DCs [205]. The TMG template design consists of the T7 promoter, sequences encoding the MHC-I signal peptide (SP), TMG^NEO^, the trafficking domain of major histocompatibility complex class I (MITD), two consecutive 3’-untranslated regions of human β-globin, and 120 adenosine poly(A) tails [69]. To generate a tandem minigene, minigenes were linked with a non-immunogenic glycine/serine linker [4,206,207]. After plasmid synthesis, in vitro-transcribed mRNA can be produced, which can be used for ex vivo loading of autologous DCs or LNP encapsulation to produce the final vaccine [208]. 

BioNTech SE has developed an iNeST platform for patient-specific cancer antigen therapy, including BNT121 and BNT122. BNT121, a vaccine containing 10 neoantigens, was tested by intranodal administration in 13 melanoma patients. It was found to induce T cell infiltration to kill tumor cells and to have recurrence-free disease activity (NCT02035956) [4]. Strong immunogenicity has also been observed in a number of tumor types following injection of BNT122 (RO7198457), which contains up to 20 patient-specific novel epitopes (NCT03289962). mRNA-4157 is another personalized mRNA cancer vaccine developed by Moderna, which contains 20 neoepitopes with strong immunogenicity selected according to the unique characteristics of the patient’s immune system and specific mutations. The mRNA is encapsulated in the LNP, and the vaccine is injected intramuscularly. The drug is being tested for an acceptable safety profile and observed clinical responses in patients with solid tumors (NCT03313778) and melanoma (NCT03897881) (Table 1).

### 4.2. TAA mRNA Vaccines

In addition to neoantigen vaccines, another class of tumor antigen vaccines is also widely used, namely tumor-associated antigen (TAA) vaccines [210,211]. TAAs are autoantigens that are preferentially or abnormally expressed in tumor cells and can also be expressed at certain levels in normal cells. They can be classified into the following categories [212,213]: (1) cancer/germline antigens (or cancer testis antigens), which are normally expressed only in immune-privileged germline cells but are transcriptionally reactivated in tumor cells (e.g., melanoma antigen gene family (MAGE), B-M antigen-1 (BAGE), New York esophageal squamous cell carcinoma (NY-ESO-1), and synovial sarcoma X chromosome breakpoint-2 (SSX-2)) [214]; (2) cell lineage differentiation antigens, which are derived from normal tissues (e.g., tyrosinase, glycoprotein 100 (gp100), melanoma antigen recognized by T cells 1 (Melan-A/MART-1), prostate-specific antigen (PSA) and prostate acid phosphatase (PAP) in prostate cancer, and mammaglobin-A (MAM-A) in breast cancer) [215]; and (3) proliferation-, differentiation-, and antiapoptosis-related proteins with tumor-selective high expression contributing to the malignant phenotype (e.g., carcinoembryonic antigen (CEA), human telomerase reverse transcriptase (hTERT), human epidermal growth factor-2/neu (HER2/Neu), baculoviral inhibitor of apoptosis repeat-containing protein 7 (livin), baculoviral inhibitor of apoptosis repeat-containing 5 (survivin), and mucin-1 (MUC-1)) [216]. Despite significant differences in the expression of TAAs in normal tissues and cancer cells, TAAs are characterized by low tumor specificity and low immunogenicity [217,218]. Therefore, cancer vaccines using these antigens must be sufficiently effective to break immune tolerance with several features. Incomplete peripheral tolerance of TAA-reactive T cells and very low expression of TAA in peripheral tissues are critical for restoring immunoreactivity via expression of the relevant TAA in APCs [219].

In 1995, the first TAA mRNA encoding the human carcinoembryonic antigen CEA was constructed, capped, polyadenylated, and stabilized by the 5′ and 3′ UTRs of human β-globin. After the injection of naked mRNA into mice, CEA antibody production was observed, which was the first proof of concept for TAA mRNA vaccines for cancer therapy [220]. A series of TAA mRNAs were then validated in a mouse cancer model, including gp100 [221], melanoma antigen recognized by T cells 1 (MART1) [222], and tyrosinase-related protein 2 (TRP2) [223,224] in B16F10 melanoma tumors, cytokeratin19 mRNA in Lewis lung cancer [225], and CD133 mRNA in gliomas [226]. 

The main problem in the development of TAA mRNA vaccines is the achievement of immunogenicity from TAA. The use of multiple shared TAA mRNA has become the main trend in the development of clinical cancer vaccines, which have been verified in various clinical trials and show strong potential for the induction of antitumor immune responses [227] (Table 2). Vaccination with DCs electroporated with mRNA encoding WT1 (NCT00965224) or WT1, PRAME, and CMVpp65 (NCT01734304) or CT7, MAGE-A3, and WT1 mRNA (NCT01995708) or WT1/PRAME (NCT02405338) mRNA was tested in acute myeloid leukemia (AML). An increase in antigen-specific T cells and induced antibody responses was observed [228,229], and overall survival (OS) improved [230]. 

Melanoma, a form of skin cancer, is a malignant tumor that is prone to metastasis. Because of the location of the lesion, which lends itself to the local injection of mRNA with a high degree of safety, melanoma mRNA vaccines have been tested in several clinical trials and have significantly advanced. BNT111, a mixture of RNA-LPX encoding four TAAs (NY-ESO-1, MAGE-A3, tyrosinase, and TPTE), has shown great therapeutic potential alone or in combination with the PD-1 inhibitor, inducing strong CD4^+^ and CD8^+^ T cell immunity and maintaining antitumor effects for months after vaccination was ceased [107]. Moderate flu-like symptoms (such as fever and chills), which were classified as grade 1-2 adverse events, occurred in 5% of the patients (NCT02410733). Based on these results, BNT111 was fast-tracked by the FDA for a phase II clinical trial with the anti-PD-1 antibody cemiplimab in patients with anti-PD-1 refractory or relapsed, unresectable stage III/IV melanoma (NCT04526899). The BNT112 cancer vaccine has also been tested as monotherapy or in combination with cemiplimab in patients with prostate cancer (NCT04382898). BNT114 (a mixture of TAA mRNAs encoding breast cancer antigens) and BNT115 (a mixture of three ovarian cancer antigen mRNAs) are being developed. Reinhard et al. described another strategy, called CarVac, in which TAA mRNA was used as a chimeric antigen receptor (CAR)-T therapy stimulator to achieve adjustable expansion of low doses of CAR-T cells. CLDN6-CAR-T cells gradually disappeared from the tumor microenvironment (TME) in the absence of a proliferation signal. Administration of CLDN6 mRNA-LPX (BNT211) effectively induced APCs to present antigens, and the number of CLDN6-CAR-T cells peaked 3-4 days after vaccination and then declined. Good safety and efficacy have also been demonstrated after multiple administrations [236].

Another mRNA drug company, CureVac AG, has developed a series of RNActive^®^ vaccines that use chemically unmodified, sequence-optimized mRNA to encode TAAs for cancer treatment [235]. Specific cytotoxic T lymphocytes and antibodies can be induced by exposure to unmodified mRNA to produce self-adjuvants. CV9103, a prostate cancer vaccine containing protamine-stabilized mRNA encoding the antigens PSA, PSCA, PSMA, and STEAP1, was well tolerated in a clinical trial of 48 participants and induced immune responses that could lead to prolonged patient survival [231] (NCT00906243, NCT00906243). CV9201 is another mRNA-based cancer immunotherapy encoding five TAAs (NY-ESO-1, MAGE-C1, MAGE-C2, survivin, and 5T4). In 60% of the patients, there was more than a twofold increase in B cells directed against antigens after treatment with CV9201 [232]. CV9202 contains mRNAs encoding six different NSCLC TAAs (MUC-1, survivin, trophoblast glycoprotein, NY-ESO-1, MAGE-C1, and MAGE-C2) (NCT01915524). Following intradermal administration, antigen-specific immune responses increased in 84% of patients; 80% of patients had a 40% increase in antigen-specific antibody levels and functional T cell levels, and 52% of patients had multiple antigen specificities [235] (NCT01915524). Based on these studies, CV9202 has also been evaluated in phase I/II studies in combination with the anti-PD-L1 antibody durvalumab or the anti-CTLA4 antibody tremelimumab, administered subcutaneously with a needle-free injection device (NCT03164772).

Standardization of TAA mRNA construction is also possible; BNT111 is a good example. The addition of a 5′-cap analog, 5′ and 3′ UTRs, and a poly(A) tail can increase mRNA stability and translation efficiency. The full-length TAA-coding sequence was tagged with a signal peptide (SP), tetanus toxoid CD4^+^ epitopes P2 and P16, and MITD for enhanced HLA presentation and immunogenicity [107] (Figure 8).

In TAA mRNA vaccines, other strategies have been used to activate antigen-presenting cells, such as the electrical transfer of DC in adoptive therapy or administration of antigen mRNA targeting the spleen. These strategies have considerable therapeutic value in AML and offer potential treatment options for non-solid cancers that are difficult to treat.

Clinical trials with an mRNA cancer vaccine have shown that vaccination against mutant epitopes or TAAs was safe and well tolerated, with most of these conditions being early onset, transient, and manageable. When injected intramuscularly, the most common adverse events of mRNA-LNP were pain at the injection site, fatigue, headache, arthritis, and myalgias [209]. When the mRNA-based cancer vaccine was administered intravenously by LPX, the clinical adverse events were mild to moderate flu-like symptoms, such as pyrexia and chills [107]. Future preclinical and clinical studies should investigate potential safety concerns such as local and systemic inflammation.

### 4.3. mRNA Encoding Ab

Since the development of hybridoma technology for the production of monoclonal antibodies (mAbs) in 1975, antibodies have become the most rapidly developing cancer-targeted drugs [237]. A series of antibodies that mediate tumor cell killing by antibody-dependent cellular cytotoxicity (ADCC), antibody-dependent cell phagocytosis (ADCP), and complement-dependent cytotoxicity (CDC) activities, or by immunosuppressive signal blockade, are well used in clinical trials. Conventional antibodies consist of antigen-binding sites (Fabs) and constant region (Fc) fragments. Fab fragments bind to tumor antigens and the Fc region lyses cancer cells by interacting with Fc receptors (FcγRs) on effector cells (such as NK cells and macrophages) [238]. Many chimeric antibodies against cancer antigens (murine Fab and human Fc regions) have been approved for clinical use (Table 3). In addition, many immune checkpoint inhibitors (ICIs) are also widely used in immunotherapies [238,239,240,241] (Table 4) and are often combined with other therapies (such as neoantigen mRNA and TAA mRNA). In addition to traditional antibodies, antibody fragments (including single-chain variable fragments (scFvs) and single-domain antibodies (sdAds)) and bispecific/multispecific antibodies have shown great potential in immunotherapies. Bispecific antibodies (bsAbs) have two antigen-binding arms and function to mediate immune cell killing by forming a T cell–bsAb–tumor cell complex, blocking two receptors of tumor cells [242]. 

However, customization of each antibody, quality control, and purification are challenges for mass antibody production. Therefore, the use of mRNA to generate intact mAbs in vivo was tested. Compared with protein antibody therapy, mRNA platforms have some unique advantages: (1) different antibodies can share the same design, production, and purification protocol of IVT mRNA; (2) optimized variants can be produced by changing the coding region of the IVT mRNA; (3) IVT mRNA uses the cells’ own ribosomes to encode proteins and undergoes correct assembly and post-translational modification; (4) as the serum half-life of the mRNA-encoded Ab is determined by the half-life of both the Ab itself and the mRNA, the half-life of short-lived proteins can be extended [243]; and (5) in the current study, there was no upper dose and no dose-limiting toxicity for antibody mRNA administration. mRNA therapy features easy quality control, rapid production, and good tolerance and safety, which makes it a better mAb protein alternative [244].

In 2008, CureVac attempted the expression of mRNA-encoded antibodies against HER2, EGFR, and CD20 in vitro. Nine years later, the CureVac team tried to use mRNA-LNPs encoding the anti-CD20 antibody rituximab in vivo and established high serum titers in mice with curative effects of significant inhibition of tumor cell growth in lymphoma models, demonstrating for the first time that mAb mRNA is effective in cancer immunotherapy [245]. Rybakova et al. tested the pharmacokinetics and pharmacodynamics of the mRNA-encoded anti-HER2 antibody, trastuzumab, and demonstrated its anticancer activity [246].

In addition to monoclonal antibodies, a series of mRNA-encoded bispecific antibodies (bsAbs) have been developed. Two chemokines, chemokine ligand 2 (CCL2) and CCL5, play major roles in the accumulation of tumor-associated macrophages (TAMs) and induction of immunosuppression in hepatocellular carcinoma (HCC). To prevent immune cell chemotaxis, a bsAb, BisCCL2/5i, which binds CCL2 and CCL5, was developed by Wang et al. The drug effectively promotes the differentiation of TAM into the antitumoral M1 phenotype and reverses immunosuppression in the TME. The use of BisCCL2/5i renders HCC sensitive to trimeric PD-1 ligand inhibitors (PD-Li) and prolongs survival in liver malignancy models [247].

Bispecific T cell engagers (BiTEs) are a class of bsAbs without the Fc region. They consist of two single-chain variable fragments (scFv) joined by a flexible linker. One scFv recognizes the T cell surface protein CD3, whereas the other scFv binds to a target antigen on cancer cells. This specific structure of BiTEs enables the localization of T cells to tumor cells and thus mediates tumor killing [248]. Stadler et al. generated a RiboMab platform with three BiTE mRNAs targeting three tumor-associated antigens (TAAs) (CD3 × tight-junction proteins claudin 6 (CLDN6), claudin 18.2 (CLDN18.2) × CD3, and epithelial cell adhesion molecule (EpCAM) × CD3). mRNA-encoded CD3×CLDN6 BiTE (which remained above half-maximum levels for up to 6 days) had a longer duration in serum than protein (which was barely detectable after 24 h). CD3×CLDN6 and EpCAM×CD3 IVT mRNA in a human ovarian cancer xenograft mouse model showed complete tumor regression without a systemic immune response [249]. CD3×CLDN6 mRNA (BNT142) is currently in phase I/II clinical trials (NCT05262530) (Table 5).

Although the number of clinical studies relying on mRNA antibody expression is still very limited, the applications of both mAb and bsAb have already been validated. Targeting cancer antigens, blocking immunosuppressive molecules on the surface of cancer cells, and mediating the antitumor effect of T cells through mRNA-encoded antibodies demonstrate the great potential of mRNA antibody immunotherapy. The development of mRNA antibody platforms is expected to lead to more optimal antibody design, longer half-life, and more clinical product applications in the future.

### 4.4. Immunomodulator mRNA Vaccines

The TME is closely associated with tumorigenesis and development. Tumor cells mediate immune suppression by releasing signaling molecules into the TME. This explains the difficulty in activating immune responses in tumors and results in the failure of cancer therapies in some patients [250]. Therefore, it is important to restore the antitumor immune response environment by regulating immunosuppression with immunomodulatory agents [251]. Clinically, injecting cytokines into cancer patients has become a cancer treatment strategy. For instance, more than 140 clinical trials have been launched to test type I interferon (IFN-I), which can directly induce apoptosis of tumor cells, prevent angiogenesis of tumor blood vessels, activate mDCs, and promote the differentiation of effector T cells [252]. Cytokines that activate antitumor effector cells (IL-12, IL-23, IL-36, GM-CSF, and IFN-α), costimulators (OX40L (CD252), inducible costimulatory ligand (ICOSLG/CD275), tumor necrosis factor receptor superfamily 9 (TNFSF9/4-1BBL/CD137L)), pattern-recognition receptor (PRR) agonists (TLRs and RIG-I agonists), and others are commonly used in immunotherapy [250]. Commonly used antitumor cytokines include interferons, interleukins, lymphokines, and tumor necrosis factors with various functions. Some have proinflammatory functions (IL-23, IL-36γ, IFN-α), stimulate the proliferation and differentiation of immune cells (CD70, IL-15, GM-CSF), or activate lymphocyte functions (IFN-γ, IL-12, IL-27). Costimulatory molecules act as stimulatory immune regulators to enhance the magnitude of immunological responses against malignant cells by binding to T cell surface receptors [253]. PRR agonists activate the innate immunity and release various cytokines to activate the immune system [254]. Current immunomodulator therapies have some clinical limitations, such as severe dose toxicity due to their short half-life, repeated administration, and systemic delivery (such as IL-12). Therefore, intratumoral (i.t.) and intradermal (i.d.) injections are commonly used to induce local immune responses. The standout advantages of both transiently induced protein expression and delivery via the local route make mRNA therapy well suited to modulate the TME, and a number of preclinical studies have been performed.

IL-12 is a well-described cytokine important for the activation of cytotoxic T lymphocytes (CTLs) and natural killer (NK) cells. In 2018, the therapeutic effect of IL-12 mRNA-LNPs on *MYC* oncogene-driven hepatocellular carcinomas (HCC) was verified [255]. In this case, the liver-targeted delivery feature of LNP was used to target HCC, but this mode of administration is not applicable to many other cancers. Then, more intratumorally (i.t.) delivered mRNA was tested in mice. Furthermore, because of the unique functions of each cytokine, the use of a single cytokine has limited effects on tumor treatment. Therefore, multiple cytokines with different functions are often combined to achieve improved therapeutic effects. The efficacy of IL-12, IL-27, GM-CSF, and their combination encapsulated in di-amino lipid nanoparticles was tested in the B16F10 model. Administration of IL-12 and IL-27 mRNA appeared to induce NK and CD8^+^ T cells in the TME and showed the best therapeutic effect [256]. Another preclinical study evaluated the intratumoral delivery of an mRNA mixture (IL-12, GM-CSF, IL-15, and IFN-α) in a B16F10/CT26 tumor model. mRNA expression increases the number of proinflammatory CD4^+^ and CD8^+^ T cells in the TME and induces an immune response in distal tumors. The addition of anti-PD-1 antibodies further improved the survival rate of the mice [257]. In 2019, Haabeth et al. established a precedent for the combination of cytokines and costimulator mRNA to initiate global anticancer immunity. They utilized a charge-altering releasable transporter mRNA delivery platform to induce the local expression of cytokines (CD70, IL-12, and IFN-γ) and costimulators (OX40L, CD80, and CD86) individually and in combination in two tumor models (A20B-cell lymphoma and CT26 colon carcinoma). Mice treated with OX40L mRNA showed complete eradication of both local and distal tumors, whereas those treated with other mRNA showed only a partial response. Furthermore, the combination of OX40L with CD80 or CD86, or OX40L with IL-12 dramatically increased both survival and tumor growth delay [258]. 

These preclinical data suggest that some cytokines and costimulatory pathway molecules can be effective strategies to revitalize T cell responses in cancer, particularly when administered in combination or in combination with immune checkpoint antibodies. In 2006, the ability of the mRNA adjuvant to enhance the effect of the TAA mRNA vaccine was evaluated in a mouse model of prostate adenocarcinoma. GM-CSF mRNA co-delivery has been found to enhance the CTL response [259]. DC-activating FLT3 ligand mRNA further enhances the immunological efficacy of naked RNA vaccines [96,260]. More mRNA adjuvants have been used in clinical studies (Table 6). One of the pioneers of mRNA adjuvants is eTheRNA AG, which contains three naked mRNA molecules (constitutively active TLR4 (caTLR4), CD40L, and CD70). It promotes the activation and maturation of DCs, ex vivo or in situ, to activate T helper cells and CTLs [261,262,263]. Administration of HPV/melanoma-associated TAA mRNA in conjunction with TriMix showed a promising clinical response without increased toxicity [264,265]. A phase I study on TriMix in breast cancer is also underway (NCT03788083). In the pipeline of Moderna, mRNA-2752, an OX40L/IL-23/IL-36γ cocktail mRNA drug, promotes tumor immune infiltration and tumor regression by inducing a broad immune response involving many DC types and lymphocytes. IL-36γ and IL-23 specifically interact to mediate antitumor efficacy, while the T cell costimulator OX40L significantly increases lymphocyte response rates. Notably, in an immunologically barren tumor mouse model (B16F10-AP3), the combination of the drug and ICIs increased survival to 85%, whereas tumor cells were insensitive to ICIs alone [266]. A dose-escalation study of mRNA-2752 in various advanced malignancies and an observational study of mRNA-2752 in combination with the anti-PD-1 antibody pembrolizumab in ductal carcinoma are also ongoing (NCT03739931, NCT02872025). Another mRNA adjuvant containing only OX40L (mRNA-2416) is also being tested for tolerability and safety in combination with the anti-PD-L1 antibody durvalumab in metastatic ovarian and lymphoma cancers (NCT03323398). Similarly, MEDI1191 (IL-12 mRNA) has also demonstrated excellent safety, tolerability, and efficacy in combination with durvalumab for the treatment of solid tumors (NCT03946800). Another IL-12 mRNA product, BNT151, developed by BioNTech, is currently in phase I testing for metastatic tumors (NCT03871348).

These studies suggest that local modulator mRNA therapy enables many immunosuppressed or immune-cell-deficient TMEs to remodel their function and elicit a global immune response from various DCs and lymphocytes, showing exciting therapeutic results in distal tumors and multidrug-resistant metastatic tumors. In particular, when combined with ICIs, they show enhanced antitumor responses. We anticipate that modulators with different functions can be used in the field of cancer treatment to advance in situ vaccination against cancers and achieve long-term benefits.

### 4.5. Protein Replacement Therapy

Tumor suppressor genes (TSGs) play important roles in maintaining genome integrity and regulating cell proliferation, differentiation, and apoptosis. The loss of function of TSGs is usually associated with cancer development, progression, and treatment resistance [270]. In addition, several human cancer exome sequencing studies have uncovered a series of cancer driver genes, most of which are TSGs [271]. Several key signaling pathways and processes are associated with the most likely cancer-driving TSGs, including the Wnt/β-catenin pathway (adenomatous polyposis coli (*APC*), *AXIN1*, and cadherin-1 (*CDH1*)), the phosphoinositide 3-kinases (PI3K)/protein kinase B (AKT)/mammalian target of rapamycin (mTOR) pathway (phosphoinositide-3-kinase regulatory subunit 1 (*PIK3R1*), phosphatase and tensin homolog (*PTEN*), and tuberous sclerosis proteins ½ (*TSC1/2*)), cell growth and differentiation (ras superfamily, hedgehog protein family), apoptosis/cell cycle (tumor protein P53 (*TP53*), RB transcriptional corepressor 1 (*RB1*)), chromatin modifications (CREB-binding protein (*CREBBP*), tet methylcytosine dioxygenase 2 (*TET2*), Wilms tumor 1 (*WT1*), and ubiquitin carboxyl-terminal hydrolase BAP1 (*BAP1*)), DNA damage repair (serine-protein kinase ATM (*ATM*), serine/threonine-protein kinase ATR (*ATR*), breast cancer 1/2 (*BRCA1/2*), DNA mismatch repair protein MLH1 (*MLH1*), and DNA mismatch repair protein MSH2/6 (*MSH2/6*)), and transcriptional regulation (transcription factor GATA-3 (*GATA3*) and runt-related transcription factor 1 (*RUNX1*)) [271,272]. Loss of function occurs in most TSGs, and the cancer phenotype is mediated by hyperactivation of the mentioned pathways. In this case, a possible therapeutic approach would be able to inhibit downstream pathways by replenishing TSGs. However, when DNA transfection is used to restore functional copies, difficulties in delivery, genome integration, and mutation risk have become major obstacles to gene therapy. mRNA has been shown to be advantageous as an alternative to genes and proteins, and several preclinical studies have been conducted.

In a 2018 study, PTEN mRNA was encapsulated in PEG-coated polymer lipid hybrid nanoparticles (NPs) and introduced into PTEN-null prostate cancer cells in vitro and in vivo. Treatment with PTEN mRNA-NPs significantly promoted cancer cell apoptosis by inhibiting the PI3K/Akt pathway, and the therapeutic effect was verified in a mouse model of prostate cancer (PCa) xenograft [159]. In 2021, the team further investigated whether PTEN mRNA-NPs restored protein expression and autophagy was induced in PTEN-null cancer cells (B16F10 melanoma and anti-PD-1 ineffective prostate cancer). In addition, combinatorial treatment with anti-PD-1 antibody resulted in upregulation of CTLs and proinflammatory cytokines (e.g., IL-6, TNF-α, TNF-β, and IFN-γ) in the TME and downregulation of myeloid-derived suppressor cells (MDSCs), which also triggered immunological memory [273]. p53, one of the most frequently altered TSGs that promote apoptosis, was also tested in mRNA therapy. Kong et al. used redox-responsive particles (PDSA added) to deliver p53 mRNA in models of hepatocellular carcinomas (HCCs) and non-small-cell lung cancers (NSCLCs) and showed an effect on tumor growth inhibition. In addition, combination therapy with the mTOR inhibitor everolimus showed the strongest therapeutic effect on in situ tumors [145]. Furthermore, the team added a CXCR4-targeted peptide to hybrid NPs to achieve selective HCC targeting and high mRNA transfection efficiency. The combination of p53 mRNA-NPs and PD-1 blockade significantly reduced bloody ascites, pleural effusions, and lung metastases and prolonged survival in HCC model mice [274]. Lung-targeting LNPs were effectively used to introduce TSC2 mRNA into TSC2-null cells and suppress the mTOR pathway, resulting in improved control of tumor cell proliferation in a mouse model of pulmonary lymphangioleiomyomatosis [275].

Although the application of TSG mRNA has not been extensively explored, this restoration strategy has been demonstrated in several mouse cancer models, demonstrating its transformative and powerful potential. Therapeutic effects have been achieved in combination with immune checkpoint blockade therapy. We look forward to further applications of TSG mRNA that will take advantage of the mRNA delivery platform and advance translational medicine.

## 5. Conclusions and Perspectives

However, the treatment of cancer is challenging. Most cancer vaccine trials have limited success rates in patients with advanced disease or refractory tumors. Most commonly, effective T cell induction activation is the main obstacle. Although tumor cells have many mutations, few are recognized by the patient’s T cells because the reactivity of tumor-antigen-specific T cells is usually limited to a few mutated epitopes. Using mRNA to express multiple neoantigen peptides or tumor-associated antigens to achieve systemic DC targeting and establish neoantigen-specific T cell immunity is one of the methods used to circumvent T cell immune tolerance [107]. In addition, immunotherapy includes ICIs or other means, such as chimeric antigen receptor T cells, which may have synergistic therapeutic effects with mRNA vaccines. 

The current challenge in mRNA-based therapeutics lies in the improvement of stability and delivery specificity. The in vivo translation efficiency and stability of mRNA could be improved by optimizing mRNA technology, which requires a better understanding of RNA biology and translation processes. For example, the optimization of UTRs will increase translation efficiency and lead to tissue-specific mRNA translation [33]. Furthermore, delivery efficacy and specificity could be further improved, for example, to achieve systemic DC targeting. Achieving organ- or cell-selective mRNA delivery is the most important challenge in biomedical engineering and nanomedicine. Various lipid nanoparticles have been developed and optimized to increase cellular uptake and endosomal escape of mRNA-LNP formulations. Other lipid nanoparticles, such as antibody-conjugated LNPs and SORT LNPs, have been modulated to selectively accumulate in the target organs. Furthermore, hybrid nanoparticles containing polymers may facilitate the controlled release of mRNA. Other delivery strategies, such as the SEND system, can also be applied for mRNA delivery [276]. 

In summary, significant technological innovations have made mRNA a new class of drug in vaccine development and other medical indications. Although mRNA medicines for cancer treatment encounter a tougher road to the clinic than mRNA vaccines against infectious diseases, we believe that advances in basic mRNA biology and delivery platforms will prove that in vitro-transcribed mRNA has the potential to revolutionize cancer therapies.

## Figures and Tables

**Figure 1 pharmaceutics-15-00622-f001:**
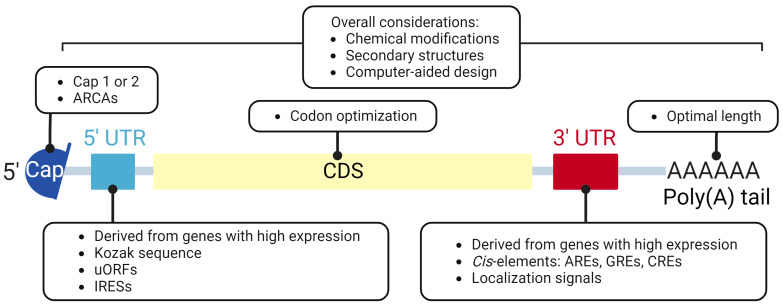
Structure and design considerations of IVT mRNAs.

**Figure 2 pharmaceutics-15-00622-f002:**
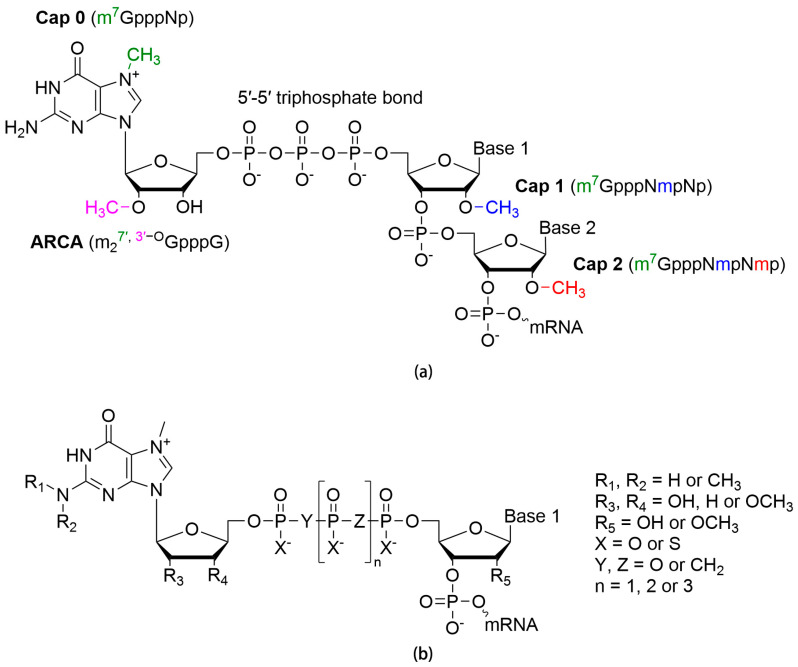
The 5′-cap structures of IVT mRNAs. (**a**) Structure of cap 0, cap 1, cap 2, and ARCA. (**b**) Chemical modifications of 5′-cap analogs.

**Figure 3 pharmaceutics-15-00622-f003:**
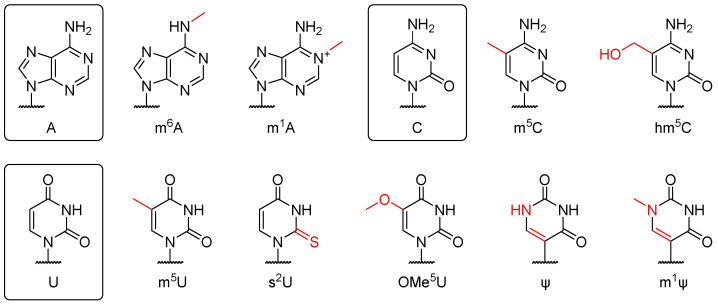
Chemical modifications on bases of IVT mRNAs.

**Figure 4 pharmaceutics-15-00622-f004:**
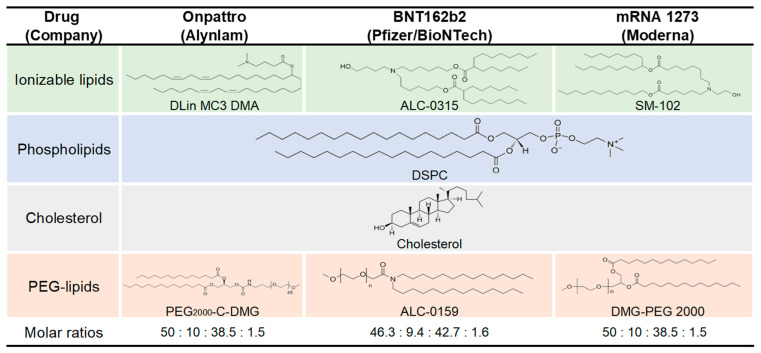
Molecular structures of lipids used in three FDA-approved clinical applications.

**Figure 5 pharmaceutics-15-00622-f005:**
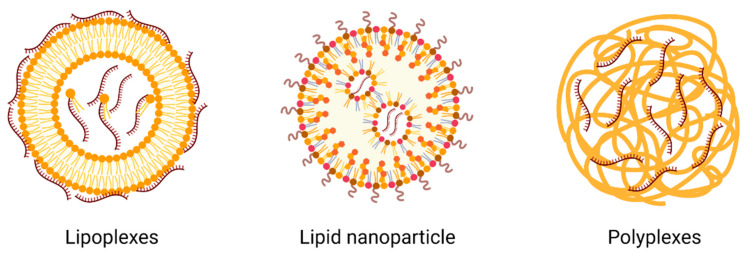
Structures of lipoplexes, lipid nanoparticles, and polyplexes.

**Figure 6 pharmaceutics-15-00622-f006:**
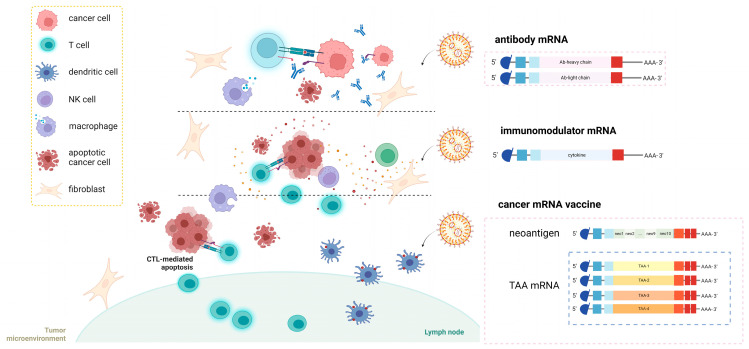
Overview of mRNA-based cancer immunotherapies.

**Figure 7 pharmaceutics-15-00622-f007:**
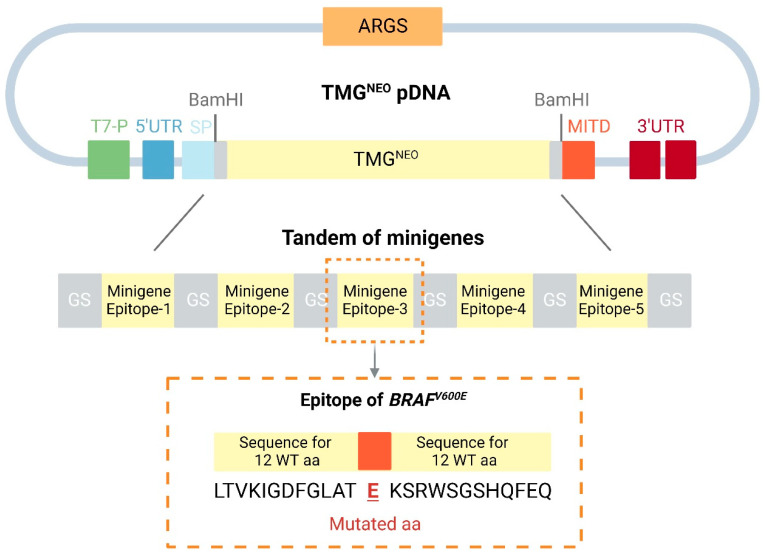
Design for TMG^NEO^ pDNA. Abbreviations: ARGS, antibiotic resistance genes; T7-P, T7 promoter; SP, signal peptide; TMG, tandem of minigenes; MITD, the trafficking domain of major histocompatibility complex class I; GS, glycine/serine linker.

**Figure 8 pharmaceutics-15-00622-f008:**
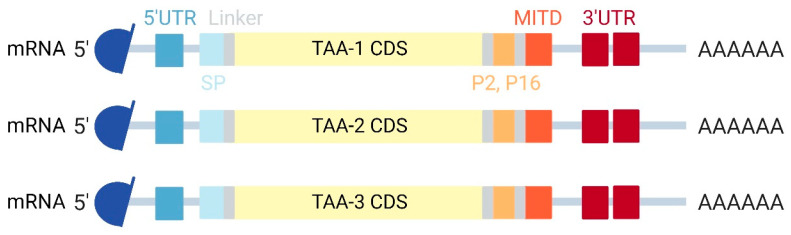
Design for TAA mRNA. Abbreviations: SP, signal peptide; P2 and P16, the tetanus toxoid CD4^+^ epitopes P2 and P16; MITD, major histocompatibility complex class I; linker, glycine/serine linker.

**Table 1 pharmaceutics-15-00622-t001:** Clinical trials of neoantigen mRNA.

Period	Product	Type	Study	Phase	Sponsor	Formulation	Route	Other Therapy	Response
2020–2023	IVAC_W_bre1_uID and IVAC_M_uID	TNBC	NCT02316457	Phase I	BioNTech SE	LPX	i.v.	/	ongoing
2017–2019	IVAC MUTANOME, RBL001/RBL002 (BNT121)	melanoma	NCT02035956	Phase I	BioNTech SE	naked mRNA	i.n.	/	not published
2017–2024	Autogene cevumeran (RO7198457, BNT122)	solid tumors	NCT03289962	Phase I	Genentech, Inc.	naked mRNA	i.v.	Atezolizumab	ongoing
2017–2025	mRNA-4157	solid tumors	NCT03313778	Phase I	ModernaTX, Inc.	LNP	i.m.	Pembrolizumab	ongoing
2018–2020	NCI-4650	solid tumors	NCT03480152	Phase I/II	National Cancer Institute (NCI)	/	i.m.	/	safe with a slight adverse event [209]
2018–2021	personalized mRNA tumor vaccine	solid tumors in digestive system	NCT03468244	NA	Changhai Hospital	LPP	s.c.	/	not published
2019–2024	RO7198457	advanced melanoma	NCT03815058	Phase II	Genentech, Inc.	LPX	i.v.	Pembrolizumab	ongoing
2019–2024	mRNA-4157	high risk of recurrence melanoma	NCT03897881	Phase II	ModernaTX, Inc.	naked mRNA	/	Pembrolizumab	ongoing
2019–2022	personalized mRNA tumor vaccine	esophageal cancer, NSCLC	NCT03908671	/	Stemirna Therapeutics	LPP	s.c.	/	not published
2019–2023	RO7198457	pancreatic cancer	NCT04161755	Phase I	Memorial Sloan Kettering Cancer Center	LPX	/	Atezolizumab, chemotherapy (mFOLFIRINOX)	ongoing
2020–2025	RO7198457	NSCLC	NCT04267237	Phase II	Hoffmann-La Roche	LPX	i.v.	Atezolizumab	withdrawn
2022–2023	SW1115C3	solid tumor	NCT05198752	Phase I	Stemirna Therapeutics	LPP	/	/	ongoing
2022–2025	neoantigen tumor vaccine	gastric cancer, esophageal cancer, and liver cancer	NCT05192460	/	Jianming Xu	/	/	PD-1/L1 drugs	ongoing
2022–2026	GRT-C901/GRT-R902	colonic neoplasms and colorectal neoplasms	NCT05456165	Phase II	Gritstone bio, Inc.	chimpanzee adenovirus	i.m	Atezolizumab, Ipilimumab, chemotherapy	ongoing
2019–2025	RO7198457	NSCLC	NCT04267237	Phase II	Hoffmann-La Roche	LPX	i.v.	Atezolizumab	ongoing
2019–2024	RO7198457	melanoma	NCT03815058	Phase II	Genentech, Inc.	LPX	i.v.	Pembrolizumab	ongoing

Abbreviations: i.v., intravenous injection; i.n., intranodal injection; i.m., intramuscular injection; s.c., subcutaneous injection; TNBC, triple-negative breast cancer; NSCLC, non-small-cell lung cancer; LPX, lipoplex; LNP, lipid nanoparticle; LPP, lipopolyplex.

**Table 2 pharmaceutics-15-00622-t002:** Clinical trials of TAA mRNA.

Period	Product/TAA	Type	Study	Phase	Sponsor	Formulation	Route	Other Therapy	Response
2022–2027	mRNA-4359 (mRNA encoding IDO and PD-L1)	advanced solid tumors	NCT05533697	Phase I/II	ModernaTX, Inc.	/	i.m.	Pembrolizumab	ongoing
2009–2013	CV9103 (mRNA encoding 4 PSAs, PSCA, PSMA, and STEAP1)	hormonal refractory prostate cancer	NCT00831467	Phase I/II	CureVac AG	protamine-stabilized mRNA	i.d.	/	well tolerated, prolonged patient survival [231]
2013–2017	CV9104 (mRNA encoding PSA, PSMA, PSCA, STEAP1, PAP, and MUC1)	PCa	NCT01817738	Phase I/II	CureVac AG	protamine-stabilized mRNA	i.d.	/	not published
2007–2009	mRNA in AML cell lysate	AML	NCT00514189	Phase I	M.D. Anderson Cancer Center	DCs loaded	i.v.	/	not published
2010–2024	tumor mRNA	PCa	NCT01197625	Phase I/II	Oslo University Hospital	DCs loaded	i.v.	/	not published
2007–2014	GRNVAC1 (mRNA encoding hTERT, LAMP)	AML	NCT00510133	Phase II	Asterias Biotherapeutics	DCs loaded	i.v.	/	not published
2011–2013	DC-006 vaccine (mRNA encoding hTERT, survivin)	recurrent epithelial OC	NCT01334047	Phase I/II	Steinar Aamdal	DCs loaded	i.d.	/	not published
2009–2012	mRNA encoding hTERT, survivin, and tumor mRNA	metastatic malignant melanoma	NCT00961844	Phase I/II	Steinar Aamdal	DCs loaded and ex vivo T cell expansion and reinfusion	i.v.	Temozolomide	not published
2009–2014	CV9201 (mRNA encoding NY-ESO-1, MAGE-C1/C2, survivin, and 5T4)	NSCLC	NCT00923312	Phase I/II	CureVac AG	protamine-stabilized mRNA	/	/	well tolerated and therapeutic [232]
2011–2023	tumor mRNA	melanoma	NCT01456104	Phase I	Memorial Sloan Kettering Cancer Center	DCs loaded	i.d.	/	ongoing
2020–2025	BNT111 (mRNA encoding NY-ESO-1, MAGE-A3, tyrosinase, and TPTE)	unresectable/stage III/stage IV melanoma	NCT04526899	Phase II	BioNTech SE	LPX	i.v.	Cemiplimab	ongoing
2019–2023	W_ova1 Vaccine (3 OC TAA mRNAs)	OC	NCT04163094	Phase I	University Medical Center Groningen	LPX	i.v.	adjuvant chemotherapy	ongoing
2020–2023	BNT112 (mRNA encoding kallikrein-2/3, acid phosphatase prostate, HOXB13, and NK3 homeobox 1)	PCa	NCT04382898	Phase I/II	BioNTech SE	LPX	i.v.	Cemiplimab	an acceptable safety profile [233]
2020–2025	BNT113 (mRNA encoding E6/E7)	unresectable/metastatic/recurrent head and neck cancer	NCT04534205	Phase II	BioNTech SE	LPX	i.v.	Pembrolizumab	ongoing
2013–2022	mRNA encoding CT7, MAGE-A3, and WT1	multiple myeloma	NCT01995708	Phase I	Memorial Sloan Kettering Cancer Center	LCs loaded	s.c.	/	safe and therapeutic with a slight adverse event [234]
2015–2019	mRNA encoding WT1 and PRAME	AML	NCT02405338	Phase I/II	Medigene AG	DCs loaded	i.d.	/	not published
2012–2018	mRNA encoding WT1, PRAME, and CMVpp65	AML	NCT01734304	Phase I/II	Ludwig-Maximilians—University of Munich	TLR7/8-matured DCs loaded	i.v.	/	feasible and safe with a slight adverse event [228]
2009–2014	mRNA encoding hTERT, survivin, and p53	breast cancer and malignant melanoma	NCT00978913	Phase I	Inge Marie Svane	DCs loaded	i.d.	Cyclophosphamide	not published
2017–2021	CV9202 (BI 1361849, mRNA encoding MUC1, survivin, NY-ESO-1, 5T4, MAGE-C1/C2)	NSCLC	NCT03164772	Phase I/II	Ludwig Institute for Cancer Research	LNP	i.d.	Durvalumab, Tremelumumab	with an adverse event
2013–2016	CV9202 (BI 1361849)	NSCLC	NCT01915524	Phase I	CureVac AG	LNP	i.d.	Radiotherapy, an EGFR tyrosine kinase inhibitor	well tolerated with an adverse event [169,235]

Abbreviations: i.v., intravenous injection; i.n., intranodal injection; i.m., intramuscular injection; s.c., subcutaneous injection; i.d., intradermal injection; PCa, prostate cancer; AML, acute myeloid leukemia; OC, ovarian cancer; NSCLC, non-small-cell lung cancer; LPX, lipoplex; LNP, lipid nanoparticle; LPP, lipopolyplex; DC, dendritic cell; LC, Langerhans cell; IDO, indoleamine 2,3-dioxygenase; PD-L1, programmed cell death 1 ligand 1; PSA, prostate-specific antigen; PSCA, prostate stem cell antigen; PSMA, prostate-specific membrane antigen; STEAP1, six-transmembrane epithelial antigen of the prostate 1; PAP, prostatic acid phosphatase; MUC1, mucin 1; hTERT, human telomerase reverse transcriptase; LAMP, lysosome-associated membrane protein; survivin, baculoviral inhibitor of apoptosis repeat-containing 5; NY-ESO-1, New York esophageal squamous cell carcinoma; MAGE-C1/C2, melanoma antigen family C1/C2; 5T4, trophoblast glycoprotein; MAGE-A3, melanoma-associated antigen 3; TPTE, putative tyrosine-protein phosphatase; HOXB13, homeobox B13; E6/E7, early protein 6/7; CT7 (MAGE-C1), melanoma-associated antigen C1; WT1, Wilms tumor 1; PRAME, preferentially antigen expressed in melanoma; CMVpp65, cytomegalovirus pp65; p53, tumor antigen p53; EGFR, epidermal growth factor receptor.

**Table 3 pharmaceutics-15-00622-t003:** Clinically approved mAbs.

Target	mAb	Type
anti-CD20 antibody	rituximab	lymphoma, chronic lymphocytic leukemia
anti-EGFR antibody	cetuximab	head and neck cancer and colorectal cancer
anti-HER2 antibody	trastuzumab	HER2-positive metastatic breast cancer

Abbreviations: EGFR, epidermal growth factor receptor; HER2, human epidermal growth factor-2.

**Table 4 pharmaceutics-15-00622-t004:** Clinically approved ICI mAbs.

Immune Checkpoint Inhibitors	Location	Ligand	mAb	Product
PD-1 inhibitors	T cells	PD-L1	Pembrolizumab	Keytruda
			Nivolumab	Opdivo
			Cemiplimab	Libtayo
PD-L1 inhibitors	Cancer cells	PD-1	Atezolizumab	Tecentriq
			Avelumab	Bavencio
			Durvalumab	Imfinzi
CTLA-4 inhibitors	T cells	CD80, CD86	Ipilimumab	Yervoy
			Tremelimumab	Imjudo

Abbreviations: PD-1, programmed death 1; PD-L1, programmed cell death 1 ligand 1; CTLA-4: cytotoxic T lymphocyte antigen 4.

**Table 5 pharmaceutics-15-00622-t005:** Clinical trials of mRNA encoding Ab.

Period	Product	Type	Study	Phase	Sponsor	Formulation	Route	Other Therapy	Response
2020–2024	BNT141 (mRNA encoding anti-Claudin18.2 monoclonal antibody)	unresectable or metastatic CLDN18.2-positive gastric, pancreatic, ovarian, and biliary tract tumors	NCT04683939	Phase I/II	BioNTech SE	LNP	i.v.	nab-paclitaxel, gemcitabine	ongoing
2022–2026	BNT142 (mRNA encoding antibodies targeting CD3 × CLDN6)	solid tumor	NCT05262530	Phase I/II	BioNTech SE	LNP	i.v.	/	ongoing

Abbreviations: i.v., intravenous injection; LNP, lipid nanoparticle; CLDN18.2, claudin 18.2; CLDN6, the tight-junction protein claudin 6.

**Table 6 pharmaceutics-15-00622-t006:** Clinical trials of immunomodulator mRNA.

Period	Product	Type	Study	Phase	Sponsor	Formulation	Route	Other Therapy	Response
2017–2023	CV8102 (mRNA encoding TLR7/8/RIG-I agonist)	advanced solid tumors	NCT03291002	Phase I	CureVac	non-coding, non-capped RNA	i.t.	anti-PD-1 Ab	well tolerated without dose-limiting toxicities [267]
2017–2019	CV8102	HCC	NCT03203005	Phase I/II	National Cancer Institute, Naples	non-coding, non-capped RNA	i.d.	Cyclophosphamide, IMA970A (multipeptide-based HCC vaccine)	safe with a side effect [268]
2016–2023	mRNA-2752 (mRNA encoding OX40L, IL-23, and IL-36γ)	high-risk DCIS	NCT02872025	Phase I	Laura Esserman	LNP	Intralesional injection	Pembrolizumab	well tolerated with slight dose-limiting toxicities [269]
2018–2023	mRNA-2752 (mRNA encoding OX40L, IL-23, and IL-36γ)	advanced malignancies	NCT03739931	Phase I	ModernaTX, Inc.	LNP	i.t.	Durvalumab	ongoing
2017–2022	mRNA-2416 (mRNA encoding OX40L)	relapsed/refractory solid tumor malignancies or lymphoma and OC	NCT03323398	Phase I/II	ModernaTX, Inc.	LNP	i.t.	Durvalumab	not published
2019–2024	BNT131 (SAR441000, mRNA encoding IL-12sc, IFNα-2b, GM-CSF, and IL-15sushi)	metastatic neoplasm	NCT03871348	Phase I	Sanofi	saline-formulated mixture	i.t.	Cemiplimab REGN2810	ongoing
2019–2027	MEDI1191 (mRNA encoding IL-12)	advanced solid tumors	NCT03946800	Phase I	MedImmune LLC	LNP	i.t.	Durvalumab	ongoing
2020–2026	BNT151 (mRNA encoding IL-2)	solid tumors	NCT04455620	Phase I/II	BioNTech SE	LPX	i.v.	/	ongoing
2021–2023	BNT152 (mRNA encoding IL-7) plus BNT153 (mRNA encoding IL-2)	solid tumor	NCT04710043	Phase I	BioNTech SE	LPX	i.v.	/	ongoing
2022–2027	ABOD2011 (mRNA encoding IL-12)	advanced solid tumors	NCT05392699	Phase I	Cancer Institute and Hospital, Chinese Academy of Medical Sciences	naked mRNA	i.t.	/	ongoing

Abbreviations: i.t., intratumoral injection; i.d., intradermal injection; i.v., intravenous injection; HCC, hepatocellular carcinoma; DCIS, ductal carcinoma in situ; OC, ovarian cancer; LNP, lipid nanoparticle; LPX, lipoplex; TLR 7/8, toll-like receptor 7/8; RIG-I, retinoic-acid-inducible gene I; OX40L, the glycoprotein OX40, OX40 ligand; IL-23, interleukin-23; IL-36γ, interleukin-36 gamma; IL-12sc, interleukin-12sc; IFNα-2b, interferon alpha2b; GM-CSF, granulocyte–macrophage colony-stimulating factor; IL-15sushi, interleukin-15sushi; IL-12, interleukin-12; IL-7, interleukin-7; IL-2, interleukin-2.

## Data Availability

Not applicable.

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
