# Peer review of "mRNA-Based Therapeutics in Cancer Treatment"

_pharmaceutics, 2023, doi:10.3390/pharmaceutics15020622_

Round 1

Reviewer 1 Report

Thank you for permitting me to review this manuscript.
This manuscript is well written. The scientific content is appropraite, mRNA vaccines are indeed a major step forward in cancer treatements appropriate explanations are given about the preparation , and  vehicle of these vaccines.  
Although authors provided many references , multiple references are missing in some statements

Line 35-40 Please provide reference (PPR)

Line 442 -475 PPR

Line 561 PPR

line 838 please rephrase as 2 consecutives in combination appears 

Line  932- 940 -953 PPR 

Please add a mini paragraph or a table mentionning the side effects described in those  valid and" in use"  mRNA  cancer  treatement to make easier for non specialist clinicians to identify these side effects or related side effects.

Please readjust tables , they are not easy to read

Author Response

Line 35-40 Please provide reference (PPR)

Response: We provided reference: Pardi N, et al., mRNA vaccines — a new era in vaccinology. Nat Rev Drug Discov . 2018 Apr;17(4):261-279. doi: 10.1038/nrd.2017.243. Epub 2018 Jan 12.

Line 442 -475 PPR

Response: We provided reference:

  1. Fu, C. and A. Jiang, Dendritic Cells and CD8 T Cell Immunity in Tumor Microenvironment. Front Immunol, 2018. 9: p. 3059.
  2. Garrido, F. and N. Aptsiauri, Cancer immune escape: MHC expression in primary tumours versus metastases. Immunology, 2019. 158(4): p. 255-266.

Line 561 PPR

Response: These descriptions we got from https://clinicaltrials.gov/, and we cited the NCT numbers.

line 838 please rephrase as 2 consecutives in combination appears 

Response: Thanks for pointing out this mistake. We rephrased sentence in the revised manuscript as follows:

“These preclinical data suggest that some cytokines and costimulatory pathway molecules can be effective strategies to revitalize T cell responses in cancer, particularly when administered in combination or in combination with other immune activating strategies, such as immune checkpoint antibodies.”

Line 932- 940 -953 PPR 
Response: We provided the reference as follows:

Sahin U, et al., An RNA vaccine drives immunity in checkpoint-inhibitor-treated melanoma. Nature. 2020 Sep;585(7823):107-112. doi: 10.1038/s41586-020-2537-9. Epub 2020 Jul 29.

AGO von Niessen, et al., Improving mRNA-Based Therapeutic Gene Delivery by Expression-Augmenting 3′ UTRs Identified by Cellular Library Screening. Mol Ther. 2019 Apr 10;27(4):824-836. doi: 10.1016/j.ymthe.2018.12.011. Epub 2018 Dec 18.

M Segel, et al., Mammalian retrovirus-like protein PEG10 packages its own mRNA and can be pseudotyped for mRNA delivery. Science. 2021 Aug 20;373(6557):882-889. doi: 10.1126/science.abg6155.

Please add a mini paragraph or a table mentioning the side effects described inthose valid and" in use" mRNA cancer treatment to make easier for non specialist clinicians to identify these side effects or related side effects.

Response: Thanks for the suggestion. In the revised manuscript, we added a mini paragraph mentioning the side effects of mRNA-based cancer treatment.  

“Clinical trials with an mRNA cancer vaccine have shown that vaccination against mutant epitopes or TAAs was safe and well tolerated, with most of these conditions being early onset, transient, and manageable. When injected intramuscularly, the most common adverse events of mRNA-LNP were pain at the injection site, fatigue, headache, arthritis, and myalgias. When the mRNA-based cancer vaccine was administered intravenously by LPX, the clinical adverse events were mild to moderate flu-like symptoms, such as pyrexia and chills. Future preclinical and clinical studies should investigate potential safety concerns such as local and systemic inflammation.”

Please readjust tables, they are not easy to read.

Response: We readjusted tables in the revised manuscript.

Reviewer 2 Report

Manuscript ID: pharmaceutics-2180268
mRNA-based therapeutics in cancer treatment
Authors: Han Sun, Yu Zhang, Ge Wang, Wen Yang, Yingjie Xu

This review article describes mRNA vaccines for cancer treatment.  It contains the mRNA-based cancer therapeutics including mRNA cancer vaccines, mRNA encoding cytokines, chimeric antigen receptors, tumor suppressors, and other combination therapies. It focuses on the molecular
design, delivery systems and clinical indications of mRNA therapies in
cancer.

This review well summarizes both mechanism and clinical trials, which are very useful not only for basic scientists but also clinicians.

I have one suggestion to improve this manuscript.

It would be nicer if the authors could describe how mRNA vesicles enter cells mechanistically.  

Author Response

It would be nicer if the authors could describe how mRNA vesicles enter cells mechanistically.  

Response: Thanks for the suggestion. In the revised manuscript, we added a mini paragraph describing how mRNA vesicles enter cells.  

“After encapsulated in a delivery vehicle, mRNA is able to enter the target cells through multiple mechanisms, which depends on the properties of the delivery platform and the cell type. For instance, mRNA delivered by lipid nanoparticles (LNP) can be internalized by micropinocytosis and endocytosis; polyplexes enter in the cells via caveolae-mediated endocytosis while lipoplexs via clathrin-mediated endocytosis or fusion with the cell membrane.”

Reviewer 3 Report

This manuscript reviewed mRNA vaccine as cancer therapeutics, discussing its advantages, current technology trend, and existing challenges. The authors first reviewed the general aspects of mRNA vaccine development, introducing the problems and corresponding solutions of sequence design (including CAP, UTRs, CDS and poly-A tail) and delivery system. Then, the manuscript focused on mRNA-based cancer immunotherapies, and reviewed neoantigen mRNA vaccine, tumor-associated antigen (TAA) vaccine, mRNA encoding Ab, Immunomodulator mRNA Vaccines, and Protein replacement therapy. The current clinical indications and pipelines of mRNA-based cancer immunotherapies are also listed.

Overall, the manuscript is clear and useful as a review. The references are well cited and discussed, and the main parts of mRNA cancer therapeutics are covered, along with the general aspect of mRNA drug development, such as the solutions to improve stability, protein expression and delivery. However, some issues need to be fixed before considering publication:

1.     In the first paragraph of introduction (lines 37-39), one of the advantages of mRNA therapeutics is described as “Second, appropriate nucleotide modifications and sequence optimization systems significantly reduce immunogenicity and improve mRNA stability and translation efficiency.” In fact, this is the disadvantage of mRNA. Because of the instability of mRNA, we HAVE TO do nucleotide modification and sequence optimization.

2.     In Figure 1, why the rectangles of UTRs are split in the middle? This is misleading because it indicates that both 5’UTR and 3’UTR are not a consecutive region.

3.     In the Chemical modification section, the authors should discuss N1- Methylpseudouridine modification, since it is the modification used in both Moderna and BioNTech’s COVID vaccines, which are the only FDA-approved mRNA vaccines.

4.     Figure 6 is too blur, expecially the legend on the top left corner.

Minor issues:

1.     Line 72: “translation efficiency (ribosome load)” is not appropriate. High ribosome load does not equal to high translation efficiency. Ribosome load is only the snapshot of ribosome binding. High ribosome load could also be due to unwanted pause, which may lead to low translation efficiency.

2.     Line 589: “TTAAs” should be “TAAs”.

Author Response

  1. In the first paragraph of introduction (lines 37-39), one of the advantages of mRNA therapeutics is described as “Second, appropriate nucleotide modifications and sequence optimization systems significantly reduce immunogenicity and improve mRNA stability and translation efficiency.” In fact, this is the disadvantage of mRNA. Because of the instability of mRNA, we HAVE TO do nucleotide modification and sequence optimization.

Response: Thanks for the advice. We revised this part as follows: “First is safety, as mRNA does not enter the nucleus; therefore, it has no risk of integration into the genome. Second, mRNA can be degraded through normal cellular pathways, and the metabolites are natural. Third, for any target protein of a known sequence, mRNA can be quickly produced in vitro by an enzymatic reaction, thereby avoiding complex manufacturing.”

  1. In Figure 1, why the rectangles of UTRs are split in the middle? This is misleading because it indicates that both 5’UTR and 3’UTR are not a consecutive region.

Response: Figure 1 has been revised to avoid misleading:

  1. In the Chemical modification section, the authors should discuss N1- Methylpseudouridine modification, since it is the modification used in both Moderna and BioNTech’s COVID vaccines, which are the only FDA-approved mRNA vaccines.

Response: N1-methylpseudouridine modification has been discussed in the revised manuscript as follows: “As the modification in two FDA-approved COVID-19 mRNA vaccines, m1ψ reduces immunogenicity compared with canonical U, with the change in mRNA structure affecting translation initiation and half-life[76-78,84]. ”

  1. Figure 6 is too blur, expecially the legend on the top left corner.

Response: We revised Figure 6.

Minor issues:

  1. Line 72: “translation efficiency (ribosome load)” is not appropriate. High ribosome load does not equal to high translation efficiency. Ribosome load is only the snapshot of ribosome binding. High ribosome load could also be due to unwanted pause, which may lead to low translation efficiency.

Response: Thanks for the advice. We removed “ribosomal load” in the revised manuscript.

  1. Line 589: “TTAAs” should be “TAAs”.

Response: We corrected it to “TAAs”.
